# Breast Cancer Screening among African Immigrants in the United States: An Integrative Review of Barriers, Facilitators, and Interventions

**DOI:** 10.3390/ijerph21081004

**Published:** 2024-07-30

**Authors:** Julian I. Rauch, Joseph Daniels, Alyssa Robillard, Rodney P. Joseph

**Affiliations:** Center for Health Promotion and Disease Prevention, Edson College of Nursing and Health Innovation, Arizona State University, 500 N 3rd St., Phoenix, AZ 85004, USArodney.joseph@asu.edu (R.P.J.)

**Keywords:** breast cancer, screening, African immigrants, United States

## Abstract

The purpose of this review was to synthesize the available literature on breast cancer-screening barriers, facilitators, and interventions among U.S. African immigrants. Following the integrative review framework and PRISMA guidelines for reporting systemic reviews, five electronic databases were searched: PubMed, CINAHL, PsycINFO, Medline, and Google Scholar. Studies were included if they were published in English language journals after 1 January 2000 and reported data on breast cancer-screening barriers, facilitators, or interventions among U.S. African immigrants. Barriers and facilitators reported by studies were descriptively examined and synthesized by two authors and classified as aligning with one of the three levels of influences based on the social–ecological model (intrapersonal, interpersonal, and community). Interventions promoting breast cancer screening were narratively summarized. Search procedures retrieved 1011 articles, with 12 meeting the criteria for inclusion in the review (6 qualitative and 6 quantitative). Intrapersonal barriers included limited awareness, fear of pain, language barriers, health concerns, transportation issues, costs, and negative past experiences. Interpersonal barriers involved modesty, spiritual beliefs, and lack of support, while community-level barriers included provider and healthcare-system challenges. Regarding facilitators, past screening experiences and health insurance were the most commonly reported intrapersonal facilitators. The only interpersonal facilitator identified was observing other women experience a breast cancer diagnosis and undergo treatment. Community-level facilitators included appointment reminders, scheduling assistance, culturally congruent interpreters, transportation to screening facilities, and patient navigators. Three articles reported outcomes of breast cancer-screening interventions. All three were pilot studies and reported increased knowledge and attitudes regarding breast cancer screening following the respective interventions. One study examined the uptake of breast cancer screening following the intervention, with results indicating an increase in screening. Findings provide a comprehensive synthesis of factors influencing breast cancer screening among African immigrants and highlight the need for future research on the topic. This review was registered with Prospero (CRD42024502826) before the initiation of search procedures.

## 1. Introduction

Breast cancer is a major public health concern. Globally, it accounts for approximately 12% of all cancer diagnoses, with over 2.3 million new cases in 2020 [1]. Breast cancer is the most common type of cancer among women and is a leading cause of cancer mortality, contributing to over 685,000 global deaths in 2020 [1,2]. In the United States (U.S.), breast cancer is the second leading cause of cancer-related death among women (behind lung and bronchus cancer), with over 310,000 new cases diagnosed annually [3]. Estimates further indicate that breast cancer will contribute to over 42,000 U.S. deaths by the end of 2024 [3].

Black women in the U.S. face significantly worse cancer-related health compared to women of other ethnic backgrounds [4]. For example, despite Black women having a lower incidence rate of breast cancer than White women (126.7 vs. 134.9 per 100,000 women), they have a 41% higher mortality rate [4]. Reasons for the high mortality rate are attributed to socioeconomic status (SES), later-stage diagnosis, and limited access to care [4,5,6]. Previous research examining breast cancer screenings among Black women in the U.S. has included limited acknowledgment of the heterogeneity that exists, including whether women were born in the US or are immigrants.

African-born immigrants differ significantly from Black/African Americans and the average U.S. population as a whole. For instance, despite having higher education and household income levels [7], African-born immigrants are more likely to experience poor physical, psychological, and social health [8,9], less likely to have a regular health care provider, and to experience a higher rate of being uninsured [10]. They also have different chronic-disease risk profiles [11] and tend to be a slightly older population [7]. In addition, African immigrants face unique barriers to healthcare that are not always universally experienced by Black Americans, including lack of English language proficiency [12], limited knowledge and mistrust of the U.S. health system [13], lack of a culturally and linguistically competent provider [14], and implicit bias among care providers regarding a patient’s immigrant status, which results in lower-quality care [15]. In addition to these barriers, emerging evidence suggests that women from certain regions of Africa (e.g., Nigeria) are at greater risk for breast cancer morbidity and mortality due to inherited genetic mutations of *BRCA1* and *BRCA2* genes that elevate breast cancer risk [16], including risk for triple-negative breast cancer, a highly aggressive form of cancer that rapidly spreads and has limited effective treatment options [17,18,19]. Given the estimated 2 million African immigrants in the U.S. [20,21], there is a pressing need to customize breast cancer-screening interventions and public health campaigns to align with their unique sociocultural norms, behaviors, and language preferences.

The purpose of this integrative review was to synthesize the available literature on the barriers and facilitators with regard to breast cancer screening among US African immigrants, as well as to review previous interventions designed to promote breast cancer screening in this population. To our knowledge, this is the first review examining these topics exclusively among African immigrants. Results provide a comprehensive synthesis of the determinants of breast cancer screening among African immigrants and describe existing evidence on strategies and interventions designed to promote breast cancer screening among African immigrants.

## 2. Methods

This integrative review followed the methodological framework proposed by Whittemore and Knafl [22]. Preferred Reporting Items for Systematic Reviews and Meta-Analysis (PRISMA) guidelines were used to report the methods and results (see Figure 1) and the review was registered in Prospero (CRD42024502826) prior to the initiation of search procedures.

### 2.1. Search Strategy

The Boolean search strategy was used to retrieve relevant articles from five electronic databases: PubMed, CINAHL, PsycINFO, Medline, and Google Scholar. Database searches included various combinations and connections of terms for two broad topical areas: study population (i.e., African immigrants) and behavior/phenomenon of interest (i.e., breast cancer screening). Search terms used for each of these topical areas are presented in Appendix A. An example of a search strategy included “breast cancer AND screening AND African immigrants AND United States OR U.S.”. Search procedures were conducted between 1 January 2000 and 31 December 2022.

### 2.2. Article Inclusion Criteria

All study designs were eligible for inclusion (i.e., quantitative, qualitative, and mixed methods), as long they met the following inclusion criteria: (1) reported data on breast cancer-screening barriers, facilitators, or interventions among African immigrants residing in the U.S., including refugees (i.e., women born in Africa that currently reside in the U.S.); (2) published between 1 January 2000 and 31 December 2022, in English language peer-reviewed journals; and (3) focused on women aged ≥18 years old. Journals that included other cancer screenings or women’s health promotion were also included in the search, as long as they reported data on breast cancer screening.

### 2.3. Study Selection Process

Figure 1 shows the article retrieval and selection process. Search procedures retrieved 1011 articles that were exported to Covidence screening and data extraction software (Cochrane Collaboration, London, UK). After the removal of duplicates (n = 241), the titles and abstracts of the remaining 770 articles were reviewed by the first author (JR) for potential relevance for inclusion in the review, resulting in 104 articles being classified as potentially eligible. Next, a full-text review of these 104 articles was independently conducted by JR and a second reviewer (RPJ) to determine article eligibility, with any discrepancies between the two reviewers being resolved by consensus. This resulted in 12 articles being identified as meeting the criteria to be included in the review.

### 2.4. Data Extraction

The following data, when applicable, were extracted from studies included in the review: (a) authors and year of publication, (b) study design, (c) sample characteristics (i.e., sample size, age of participants, race/ethnicity), (d) data collection method, (e) study purpose, (f) barriers to breast cancer screening, (g) facilitators of breast cancer screening, (h) intervention outcomes, and (i) study author reported strengths, limitations, and recommendations for future research and practice. Initial data extraction was conducted by 1 member of the research team (JR). After the initial extraction of all study data, a second member of the research team (RPJ) verified that the data extracted were accurate by comparing the data tables to the data published in each article. If there were any disagreements between these, the two reviewers reached agreement by consensus.

### 2.5. Data Synthesis

Barriers to and facilitators of breast cancer screening were descriptively examined and synthesized by the authors (JR and RPJ) using an iterative process described by Whittemore and Knafl [22]. In the first level of analysis, data extracted for studies were classified into two categories: barriers to breast cancer screening and facilitators of breast cancer screening. Next, using a constant comparison approach, the authors independently and collaboratively reviewed data extracted from studies to identify overarching themes and patterns. The themes and patterns emerging from this analysis of barriers and facilitators with respect to breast cancer screening were categorized as aligning with one of three levels of influence based on the social–ecological model: intrapersonal, interpersonal, or community [23,24]. This process of analysis occurred over a series of months and included numerous meetings between the two authors (JR and RPJ), with final decisions regarding the classification of barriers and facilitators based on the level of influences achieved by author consensus.

A narrative review approach was used to summarize each intervention included in the review. Topics included in this description included the name of the program, sample population, intervention design aim, and study outcomes. During the final phase of analysis, all authors (JR, RPJ, AR and JD) collaborated on reviewing and synthesizing the paper, ensuring that each author’s perspective was considered. Any disagreements were resolved through consensus among the authors.

## 3. Results

### 3.1. Overview of Studies

Among the 12 studies included in the review, half reported data from qualitative study designs (Table 1) and the other half from quantitative designs (Table 2). The year of article publication ranged from 2010 to 2022, with 2015 being the modal year of publication. The majority of qualitative studies employed focus group methodology [14,25,26,27], one study used one-on-one interviews [28], and one study included both interviews and focus groups [29]. Among the six quantitative studies, three reported data from cross-sectional surveys [30,31,32], one study reported cancer screening uptake following a cancer screening intervention [33], and two studies reported the feasibility and psychosocial outcomes (i.e., attitudes and knowledge) of brief, one-time education interventions designed to promote cancer screenings [34,35].

Sample sizes ranged from 14 to 38 for qualitative studies and 20 to 200 for quantitative studies. Most studies (n = nine studies) were comprised exclusively of African women [14,25,26,27,29,30,31,34,35]. Two studies also included immigrant women from non-African regions (i.e., Europe and Asia) [28,33], and one study included African American women [30]. Studies predominately included women from the African country of Somalia [25,26,27,28,31,33,34,35]. Seventy-five percent of the studies (n = 9) were conducted among women residing on the East Coast. Four were conducted in Minnesota [26,27,31,35], three in Washington D.C [14,29,32], one in Massachusetts [28], and one in New Hampshire [34]. The remaining three studies were conducted in Texas, Washington, and Kentucky [25,30,33]. Among the five studies that provided sufficient data for the calculation of the mean age of study participants [28,30,31,34,35], the mean age was 45.4 years (n = 338 participants). Only two studies reported the income levels of study participants. One of these studies was primarily comprised of women (i.e., 92%) with household incomes of less than USD 35,000/year [34]; the second study included women with mostly middle-to-upper-class incomes (i.e., 40% had income levels of USD 40,000-USD 74,999, 26% had incomes of USD 75,000–USD 119,000, and 17% had incomes ≥USD 120,000) [29]. Eight studies (66.6%) reported information on the education levels of participants. Among these, five studies predominantly included women with a high school-education level or lower [26,28,31,34,35]. Two studies reported that the majority of participants held a college or graduate degree [29,30].

### 3.2. Barriers to Breast Cancer Screening

Table 3 provides an overview of breast cancer-screening barriers reported by studies according to the level of influence based on the social–ecological model.

#### 3.2.1. Intrapersonal Barriers

**Lack of Awareness/Limited Knowledge about Breast Cancer Screening.** Lack of awareness and/or limited knowledge of breast cancer screening was one of the most frequently cited intrapersonal barriers (n = seven studies). Qualitative studies reported that participants had limited or no knowledge or exposure to breast cancer screenings before moving to the U.S. [14,25,26,28,29]. This barrier was even reported among women indicating that they regularly saw a healthcare provider in their home country [25]. As a result, the concept of engaging in breast cancer screening was relatively new, and a topic many women had only become familiar with since immigrating to the U.S. One study further reported that because of this lack of knowledge, participants were uncomfortable with receiving a breast cancer screening [26]. 

Among the two quantitative studies exploring this barrier, one cross-sectionally examined the relationship between breast cancer-screening knowledge and endorsement of receiving a screening [32]. Results showed that women with greater levels of breast cancer-screening knowledge were more likely to endorse receiving a mammography than women with lower knowledge about breast cancer screening. The second study examined knowledge of mammography delivering a breast cancer-screening intervention [34]. Findings indicated that 33% of participants lacked knowledge of a mammography before receiving the intervention; however, the specific types of knowledge assessed by the study were not reported by the authors. 

**Fear.** Fear of experiencing pain during breast cancer-screening procedures and/or following breast cancer diagnosis was another common theme across qualitative studies. Four qualitative studies reported fear of experiencing pain during breast cancer-screening procedures as a barrier to screening (i.e., pressure during mammography was a barrier in seeking screening) [25,27,28,29]. Additionally, two qualitative studies also reported that fear of being diagnosed with breast cancer was a limiting factor for undergoing screening procedures [28,29]. No quantitative studies reported examining this barrier. 

**Language.** The inability to speak the English language was a barrier cited by three qualitative studies [14,25,27] and one quantitative study [32]. Two qualitative studies indicated that participants’ inability or limited ability to speak English was a barrier to breast cancer screening, due to information materials only being available in English, as opposed to their native language [14,25]. Participants in the other study further elaborated on this topic by stating that communication with their healthcare provider relied on the use of translators, which women perceived as leading to misunderstandings between themselves and their provider [27], thus limiting their motivation to seek breast cancer screening. The only quantitative study exploring this barrier cross-sectionally examined associations between English-language proficiency and endorsement of engaging in breast cancer screening [32]. Results showed that women who reported speaking English as a primary language were more likely to report (*p* = 0.003) endorsement of engaging in a breast cancer screening than women with limited English proficiency.

**Lack of Transportation**. Lack of transportation to a screening facility was a barrier reported in two qualitative studies [25,29]. The results of these studies indicated that participants were interested in participating in breast cancer screening but lacked transportation to the screening site. Lack of transportation was also explored as a potential barrier in one quantitative study [33]. Results showed that although participants reported lack of transportation as a barrier to breast cancer screening, it was not significantly associated (*p* > 0.05) with breast cancer-screening engagement. 

**Competing Priorities.** Two qualitative studies reported the prioritization of work commitments over breast cancer screening as being a barrier [28,29]. One study further reported that participants would engage in breast cancer screening if they had a more flexible work schedule and/or if they had assistance with caring for their children [28]. 

**Health Concerns Related to Mammography.** Two studies reported health concerns associated with undergoing mammography procedures as a barrier to breast cancer screening [26,27]. Participants in one study expressed the belief that the mammography machine would cause damage to the breast, although the specific type of damage to their breasts was not reported [26]. The second study reported that some women were concerned that radiation exposure during screening procedures would cause cancer [27]. 

**Costs Associated with Breast Cancer Screenings.** One study reported the cost associated with breast cancer screenings as a barrier to engaging in screening [29]. Participants stated that the cost of breast cancer screening prevented them from engaging in the procedure as they did not have health insurance. 

**Past Negative Experience with Testing.** One qualitative study reported prior negative experiences with breast cancer screening as a barrier to future screenings [26]. However, the context related to these negative experiences was not elaborated on by the authors of this study. 

#### 3.2.2. Interpersonal Barriers

**Sociocultural Norms Related to Modesty and Privacy.** Beliefs surrounding the concepts of modesty and privacy were explored as a barrier to breast cancer screening by six qualitative studies [14,25,26,27,28,29] and one quantitative study [30]. Qualitative studies reported that participants were uncomfortable with showing their breasts and viewed it as inappropriate to have someone other than their husband touch their breasts, including healthcare providers [26,28,29]. One study further reported that some participants indicated that they would need to receive spousal approval prior to engaging in breast cancer screening [29]. Two studies also reported that participants felt uncomfortable touching their breasts when conducting a self-breast examination [14,29]. Results of the one quantitative study [35] examining the influence of modesty/privacy on breast cancer screening did not confirm these qualitative study findings, as 83% of participants enrolled in this study did not view modesty as a barrier to breast cancer screening before receiving a brief breast cancer-screening intervention. Following intervention delivery, the percentage increased to 90%. 

Norms associated with not discussing health issues with others, including preventive behaviors like breast cancer screening, were reported as a barrier to breast cancer screening in two studies [14,25]. One study reported that the women did not discuss breast cancer screening in their home country, a practice that continued even after immigrating to the US [25]. The second study reported the belief that women are expected to keep personal details related to their health, such as breast cancer status, private [14].

**Spiritual Beliefs and Religious Practices.** Spiritual beliefs and religious practices were reported as impediments to breast cancer screening in multiple qualitative and quantitative studies [14,26,27,28,30,35]. In three qualitative studies, participants expressed the fact that breast cancer screening conflicted with their religious beliefs [14,26,29]. For instance, participants believed that entrusting their lives to God would shield them from breast cancer, leading to a reluctance to undergo screening [29]. Additionally, two studies revealed conflicts between breast cancer screenings and the Muslim faith [14,26]. Results of one of these studies showed that participants perceived breast cancer screenings as an attempt to predict the will of Allah [26]. The other study reported that the act of self-examination during breast cancer screenings contradicted religious beliefs among participants [14]. Interestingly, results from the two quantitative studies examining religious or spiritual beliefs as barriers to breast cancer screening contradicted the majority of qualitative study findings. In one study, 90% of participants reported that engaging in breast cancer screenings was not perceived as conflicting with their religious beliefs [35]. The second study cross-sectionally examined the association between beliefs and breast cancer screenings, with results showing a non-significant association between religiosity and mammography intake [30].

Fatalism also emerged as a deterrent to breast cancer screening in qualitative studies [28,29,30]. Participants in one study expressed the belief that death was inevitable following a breast cancer diagnosis, undermining the effectiveness of screening [29]. Similarly, participants in another study relinquished the status of their health to God, indicating a fatalistic perspective [28]. However, findings from the one quantitative study examining fatalism as a barrier reported that 85% of women who underwent breast cancer screening disagreed with the notion that little could be done to reduce the risk of breast cancer.

**Lack of Social Support.** One cross-sectional study reported a lack of social support as a barrier to not engaging in breast cancer screening [32]. In this study, researchers stated that married women or women who live with a life partner (a proxy of social support) were more likely to endorse breast cancer screening.

#### 3.2.3. Community-Level Barriers

**Lack of Culturally Congruent Providers.** Three qualitative studies reported a lack of culturally congruent providers as a barrier to breast cancer screening. Two of these studies reported a lack of healthcare providers sharing their religion as a barrier to breast cancer screening [25,27]. The other study reported that women did not engage in breast cancer screening because their providers were not familiar with their cultural beliefs [14]. However, the authors did not describe the type of familiarity needed for women to feel comfortable with seeking breast cancer screenings from a provider. 

**Complex Nature of the U.S. Healthcare System**. Two qualitative studies reported that the complex process of scheduling and attending appointments with healthcare providers is a barrier to breast cancer screening [14,28]. One study reported that making an appointment to see a doctor was a strange concept for many women, as they did not need to schedule an appointment in advance to see a healthcare provider in their home country. Instead, they could go to a hospital or healthcare clinic and receive immediate care for a health concern [28]. Similarly, the findings of the other study indicated that navigating US healthcare was an unfamiliar and difficult process [14]. 

**Lack of Access to Breast Cancer-Screening Providers.** The lack of access to a healthcare professional or facility to engage in breast cancer screening was examined by two quantitative studies [31,32]. One study cross-sectionally examined factors associated with endorsement of breast cancer screening. Results showed that women without health insurance were 18% less likely to endorse engaging in breast cancer screening than women with insurance [32]. The second study examining this barrier reported that the majority (77%) of the participants enrolled in their study did not view lack of access as a barrier to engaging in breast cancer screening [31].

### 3.3. Facilitators of Breast Cancer Screening

Table 4 provides an overview of facilitators for breast cancer screening classified according to the level of influence based on the social–ecological model.

#### 3.3.1. Intrapersonal Facilitators

**Previous Breast Cancer-Screening Experience**. One study reported previous breast cancer-screening engagement as a facilitator for seeking additional breast cancer screenings [29]. Participants in this qualitative study indicated that previously undergoing a breast cancer screening had a positive impact on their willingness to engage in future screenings. 

**Health Insurance.** One qualitative study reported that having health insurance served as a facilitator for breast cancer screening [28]. Findings suggested that participants with health insurance coverage in this study were more likely to engage in breast cancer screenings than those without insurance. No other studies included in the review reported on the influence of health insurance coverage on breast cancer-screening behaviors.

#### 3.3.2. Interpersonal Facilitators

**Observing Others’ Experience of Breast Cancer.** One study reported that observing community members receive a breast cancer diagnosis is a motivating factor for individuals to engage in breast cancer screening [29]. Participants in this qualitative study articulated that the occurrence of one or multiple cancer deaths within the community heightened their awareness and served as motivation to undergo screening.

#### 3.3.3. Community-Level Facilitators

**Appointment Reminders and Scheduling Assistance.** Appointment reminders provided by physicians and/or healthcare providers were reported as facilitators for breast cancer-screening engagement among participants enrolled in two qualitative studies [28,29]. In addition to providing appointment reminders, one study reported that provider-based assistance with scheduling breast cancer-screening appointments further facilitated breast cancer-screening engagement [28]. 

**Interpreters with the Same Culture during Healthcare Visits.** One qualitative study reported that the utilization of interpreters during healthcare visits played a crucial role in facilitating access to preventive health screenings, specifically mentioning breast cancer screenings [28]. No other studies examined this potential facilitator of breast cancer screening. 

**Transportation to Breast Cancer Screenings.** Offering taxi vouchers and providing free transportation were recognized as facilitators of breast cancer screening in one qualitative study [28]. In the study, women were offered free transportation and taxi vouchers, addressing the transportation barrier, and promoting access to screening centers. This stands in contrast to findings from other studies where the lack of transportation was identified as a deterrent to engaging in breast cancer screening [25,29,31].

**Patient Navigators.** In a qualitative study by Saadi et al. [28], the utilization of patient navigators emerged as a significant facilitator for engaging in breast cancer screening. The study emphasized the crucial role played by patient navigators in facilitating access to preventive care services, particularly breast cancer screening. A participant in the study highlighted the indispensable support provided by patient navigators, underscoring their pivotal role in assisting individuals through the process of accessing and participating in breast cancer-screening services.

### 3.4. Review of Breast Cancer-Screening Interventions (n = 3)

**Building Better Bridges Program.** The Building Bridges program, adapted from the National Cancer Institute’s Research-Tested Intervention Programs (RTIPs), was a program that used a community health worker model to provide culturally and linguistically appropriate cancer prevention education to U.S. refugees from six cultural groups (i.e., Myanmar, the Central African Region, Bhutan, Somalia, and Arabic Speaking Countries) [33]. Specific cancers included in the program included breast, cervical, liver, and colorectal. Cultural and linguistic adaptations of the program were performed collaboratively with community health workers and community leaders. The intervention was designed to address the language preferences, culture, learning styles, and literacy levels of participants. Educational tools used to deliver the program included videos, presentations, and anatomic models. Following the delivery of the program from 2014 to 2018, screening completion data were abstracted from an appointment tracking form maintained by community health workers. Results showed breast cancer screening increased by 50% or more among most cultural groups, including Somali women. Women from Central Africa were the only participants not to reach this 50% threshold, with a breast cancer-screening rate of 35%. 

**Religiously Tailored Workshops to Increase Cancer Screening.** Pratt et al. [35] examined the feasibility of religiously tailored workshops designed to promote breast- and cervical-cancer screenings among Somali American Muslim (n = 30) women. The intervention included a single in-person workshop held at a local mosque, lasting approximately 3 h. The intervention was delivered through the use of three short videos (5–7 min in length) and facilitated discussions. The development of the intervention followed a community-engaged process, involving collaboration between community partners (including an Iman and members of a local mosque) and the research team. Workshops included time for food, tea, and prayer, and focused on leveraging Islamic values related to promoting balance among the mind, body, and spirit. Specific barriers addressed by intervention included faith practice, modesty, predestination, and preventive health care. Results showed that 97% (29/30) of participants reported that the workshop was enjoyable and 100% indicated that they would recommend the workshop to others. Additionally, the number of women reporting intent to engage in mammogram or pap smear screening within the next year increased from 80% to 93% following the intervention (*p* = 0.13). However, actual engagement in breast cancer screening was not evaluated. 

**UJAMBO program.** Piwowarczyk et al. [34] conducted a pilot evaluation of the UJAMBO program (UJAMBO is a Swahili term for greetings, good state of health, improvements, and well-being), which aimed to encourage the utilization of preventive health screenings, including mammography, pap smears, and mental health services, among African immigrant women. Twenty-one workshops were organized by community-based organizations (CBOs) in the Somali and Congolese communities at community sites. The intervention comprised a series of group workshops lasting approximately 2 h, accommodating 4–12 participants. These single-session group workshops revolved around a DVD that provided basic information about health services such as mammography and normative change by sharing stories of African immigrant women’s experiences with these services. CBO staff received training to implement the workshop using the UJAMBO DVD and guidebook developed collaboratively by the academic/community team, based on formative research in these same communities. Following the intervention, participants reported significant improvements in breast cancer-screening knowledge (*p* <.001) and intention to undergo mammography screening in the next 3 months (*p* < 0.001). Breast cancer screening following the intervention was not evaluated.

## 4. Discussion

This review provides a detailed overview of existing evidence regarding social and contextual factors influencing breast cancer-screening engagement among African immigrants, as well as interventions designed to promote breast cancer screening among this population. Findings highlight key aspects for researchers to consider when promoting breast cancer screening among U.S. African immigrants. 

### 4.1. Barriers to Breast Cancer Screening

Intrapersonal barriers to breast cancer screening were diverse, with lack of awareness and/or limited knowledge about breast cancer screenings being the most frequently cited barrier across studies, followed by fear associated with pain of screening or breast cancer diagnosis, and lack of English language proficiency. Lack of awareness and/or limited knowledge of breast cancer screening and fear are commonly reported barriers to breast cancer screening among women, regardless of race/ethnicity or immigrant status [36,37,38]. Accordingly, such concerns appear to be universally shared among immigrant women, rather than specific barriers experienced by African immigrants. Similarly, several studies investigating breast cancer screening among US immigrants from Latin countries where English is not their primary language, have also consistently reported limited English proficiency as a barrier [12,39]. This finding underscores the importance of language accessibility in healthcare settings to promote equitable access to health screenings and treatment. Barriers of limited transportation, health concerns, financial costs, and negative past experiences are also not unique to African immigrants, as these barriers have been reported extensively in the breast cancer-screening literature, regardless of race/ethnicity or immigrant status of study populations [40,41].

At the interpersonal level, modesty and privacy concerns were identified as screening barriers among qualitative studies (n = 6). Participants expressed discomfort with the physical aspects of screenings, perceiving the process as immodest or conflicting with their religious practices [14,25,26,27,28,29]. Similar findings have been observed among other ethnic groups, including Asians and Koreans [42,43,44,45,46]. An interesting finding of this review was that the sole quantitative study [35] examining modesty as a barrier to screening failed to confirm qualitative study findings, as 90% of participants in this study did not consider modesty as a barrier to breast cancer screening. The limited quantitative research on this topic highlights the need for additional larger-scale quantitative studies to examine this barrier, to expand upon previous qualitative findings. Nonetheless, tailoring breast cancer-screening interventions to address modesty concerns may represent a promising intervention design strategy to promote breast cancer screening among African immigrants. 

Conflicts stemming from religious or spiritual beliefs were reported across qualitative studies. Several studies reported that participants viewed breast cancer screening as unnecessary, believing that divine intervention protected them from breast cancer [14,26,29]. In addition, two studies highlighted conflicts between breast cancer screenings and the Muslim faith [14,26], with participants perceiving screening as an attempt to predict Allah’s will [26]. Self-breast examination was also considered inconsistent with religious beliefs [14]. Two quantitative studies, however, did not entirely endorse the notion that religion posed a barrier. For example, in one study, 90% of participants did not perceive religious beliefs as impeding breast cancer screening before the intervention [35]. Similarly, in the second study, there was no notable association found between religion and mammography [30].

Fatalistic views were also prevalent among participants in several qualitative studies, owing to the belief that Allah predetermined their breast cancer fate, potentially complicating screening decisions [27,28]. Interestingly, quantitative findings contradicted these beliefs, with the majority perceiving fatalism as unrelated to screening [30]. Despite discrepancies, addressing fatalism in breast cancer interventions for African immigrants represents a critical opportunity for culturally tailoring breast cancer-screening interventions, as leveraging religious beliefs to promote behavior change has a longstanding history for improving health outcomes [47].

Norms associated with not discussing health issues with others, including preventive behaviors like breast cancer screening, were reported as a barrier in two studies [14,25]. One study emphasized that women refrained from discussing breast cancer screening in their home country because cultural norms forbid women from speaking negatively about their bodies [25]. This silence resonated with the findings of another study [14], where women avoided discussing breast cancer screening due to it being perceived as taboo. The latter study also reported the belief that women are expected to keep details related to their health, such as breast cancer status, private [14]. Understanding and addressing these privacy concerns are crucial for designing culturally sensitive interventions that can effectively raise awareness, bridge language gaps, alleviate fears, and address logistical and financial barriers within this demographic.

In contrast to the numerous individual and interpersonal barriers identified in our review, only a limited number of studies (n = three) explored community-level barriers to breast cancer screening. The most frequently cited barriers included the lack of culturally aligned providers [14,25,27] and the complexity of the U.S. healthcare system [14,28]. Qualitative studies reported that women felt disconnected from healthcare providers who did not share or understand their religious beliefs, contributing to a sense of alienation. Participants in several studies also expressed challenges in scheduling healthcare appointments, which contrasts with the walk-in basis of healthcare services in their home countries [14,28]. These collective findings underscore the ongoing need for cultural humility and competency training for healthcare providers. Such training has the potential to assist healthcare providers in bridging cultural gaps, fostering a more inclusive and understanding healthcare environment, and alleviating the fears and concerns that ethnic women may experience during screening interactions. Similarly, understanding the cultural and social differences in the health care systems among African countries and designing culturally tailored programs to address these issues represents another leveraging point for culturally tailored breast cancer-screening interventions.

Studies examining the role of health insurance on breast cancer-screening behaviors reported somewhat inconsistent results. Among the two quantitative studies exploring this barrier, one reported that uninsured women were 18% less likely to engage in breast cancer screening [32]. The second study reported that 77% of participants did not perceive lack of access as a hindrance [31]; however, we hypothesize this outcome may have been related to the high percentage of participants who reported having insurance coverage. Regardless of the limited research on this topic among African immigrant women, findings from the broader literature underscore insurance’s pivotal role in healthcare utilization, including promoting breast cancer-screening engagement [48,49,50].

### 4.2. Facilitators of Breast Cancer Screening

A notable finding of this review was the limited number of studies including a focus on facilitators of breast cancer screening (compared to the number of studies examining barriers to screening). At the intrapersonal level, only two facilitators were identified across studies: previous positive experiences with undergoing a breast cancer-screening intervention and having health insurance coverage. Ndukwe et al. [29] reported that women with prior screening experience were more inclined to participate in future screenings. Participants of this study were women who initially avoided screening due to negative perceptions of breast cancer screening, which were a result of the influence of normative referents within their community (i.e., family members and friends, suggesting a belief that breast cancer screening was a painful process). The findings of this study also found that previously experiencing a breast cancer screening served as a motivational factor to actively engage in future breast cancer screenings [29]. This finding aligns with the theoretical construct of mastery experiences for the enhancement of self-efficacy [51], which is also an evidence-based behavior-change technique for promoting health screenings [52], underscoring the importance of positive first-hand experiences with breast cancer screening.

Although also explored as a barrier in several studies, health insurance coverage was identified as a facilitator in one study [31]. Findings of this study showed that participants reported that women with health insurance were more likely to undergo breast cancer screenings, although it is important to note that almost all participants in this study had insurance coverage. Regardless of whether health insurance is conceptualized as a barrier to or facilitator for breast cancer screening, the extant literature supports the notion that insurance coverage is positively related to both screening behaviors and health outcomes [53,54]. Ensuring accessible free or affordable health insurance is crucial, especially for low-income or underserved populations like immigrants, not only for heightened access to preventive services like breast cancer screening but also for overall disease management and potentially improved health outcomes in these communities.

At the interpersonal level, the only facilitator for breast cancer screening identified across studies was observing other community members experience breast cancer [29]. This finding is consistent with the concept of observational or social learning, a seminal phenomenon in Bandura’s Social Cognitive Theory, (SCT), which posits that learning occurs through social observation and subsequent imitation of modeled behavior [55]. This approach has been identified as an evidence-based strategy for promoting healthcare screenings across the study population [56,57,58]. As such, investigators developing future breast cancer-screening interventions for African immigrants should identify effective design strategies to leverage this concept when promoting breast cancer screening among African immigrants (e.g., incorporating testimonials from breast cancer survivors describing their diagnosis/treatment experience and encouraging breast cancer screening).

Facilitators for breast cancer screening identified at the community level encompassed various strategies to encourage screening, including appointment reminders and cultural support during healthcare visits [28,29]. These findings are consistent with other reviews indicating that client reminders effectively facilitate breast cancer, cervical, and colorectal screenings [59,60]. Similarly, employing interpreters from the same culture during healthcare visits, utilizing patient navigators, and providing transportation to breast cancer screenings are also established practices for improving healthcare delivery and outcomes among immigrant and other underserved populations [28,38,61,62,63,64]. Patient navigation programs, in particular, have an established effectiveness for increasing breast cancer screenings in various cancer screenings, including breast cancer [62,64], as does providing free transportation to screening facilities [38].

### 4.3. Breast Cancer-Screening Interventions

The results of this review also highlight the limited number of culturally tailored breast cancer-screening interventions for African immigrants. Among the three identified studies, two were pilot studies evaluating the preliminary effects of brief, one-time interventions on breast cancer-screening knowledge or intention [34,35]. The results of these studies showed enhanced knowledge and intention to engage in breast cancer screening. Only one study examined the actual uptake of breast cancer screening following an intervention [33], and reported a marked increase in breast cancer screening following the delivery of the intervention. Despite the limited number of intervention studies promoting breast cancer screening, several common themes across the three intervention studies reviewed were observed. All interventions included some type of tailoring designed to address the cultural and linguistic needs of the African immigrant communities they targeted. Strategies reported included engaging community health workers (CHWs), collaborating with community leaders, and adapting education tools to address language preferences, cultural norms, religious beliefs, and literacy levels of the participants. Numerous studies have indicated favorable outcomes through the implementation of culturally tailored interventions. For instance, a narrative review conducted by Racine and Andsoy [63] demonstrated improved mammography rates among Muslim refugees through the delivery of culturally tailored interventions. Similarly, Allen and Bazargan-Hejazi [65] evaluated the effectiveness of a culturally and ethnically tailored telephone interventions to enhance mammography screening among women, and reported that the intervention resulted in an 8% increase in mammography rates. Collaboration between community-based organizations (CBOs) and local community members played a vital role in shaping the intervention content, ensuring it resonated with the needs and preferences of the specific population being targeted.

The use of diverse educational tools to convey health information was also a theme observed across all three interventions. These tools included videos, presentations, anatomic models, and a DVD, and aimed to enhance understanding and engagement among participants. Given the findings of all three of these interventions demonstrated positive outcomes regarding acceptability and feasibility, researchers should consider incorporating and building upon such methods of intervention delivery in future breast cancer-screening interventions for African immigrants.

An interesting finding of our review was that only one of the interventions reviewed appeared to be based on established theories or models of behavior change. This study, conducted by Pratt et al. [35], leveraged both the social–ecological model and the behavior-change strategies of Social Cognitive Theory by incorporating behavior, cognition, and environment to influence health behaviors or outcomes. Given the evidence supporting the notion that theoretically based behavior-change interventions for breast cancer screening are more effective than those that are atheoretical [66,67], future researchers are encouraged to leverage existing evidence-based behavior-change strategies and theories when designing breast cancer-screening interventions for African immigrants.

We were also surprised to find that the interventions reviewed did not appear to address many of the screening barriers identified in our review. Although a few of the barriers identified were addressed (i.e., modesty and faith [35] and knowledge [34]), many were not (i.e., transportation to screening facilities, appointment reminders, and scheduling). This lack of direct attention to identified barriers suggests potential areas for enhancement in future breast cancer-screening interventions.

### 4.4. Strengths and Limitations

The current review offers several strengths. First, to our knowledge, it is the first review to comprehensively examine barriers, facilitators, and interventions with respect to breast cancer screening among African immigrants in the U.S. This unique focus contributes to a comprehensive understanding of the breast cancer-screening landscape within this demographic. Second, the integrative methodology employed enables the incorporation of diverse study types, ensuring a thorough examination of the literature. The systematic cross-case data analysis employed facilitates the identification of overarching themes, providing nuanced insights into factors influencing breast cancer screening [22]. Adherence to PRISMA guidelines also enhances the transparency and credibility of the findings reported. Lastly, the application of the social–ecological model enriches understanding by elucidating the multifaceted influences shaping breast cancer-screening behavior among African immigrants [23,24].

Several limitations of our review should also be noted. The over-representation of Somali immigrants from the East Coast in the included studies may limit generalizability to other African immigrant groups and regions. Additionally, incomplete reporting of participant sociodemographic characteristics hampers the ability to draw broad conclusions. Categorizing barriers and facilitators within a single level of influence under the social–ecological model, while strengthening the providing of a holistic framework, may oversimplify nuanced influences reported in the literature. Furthermore, potential publication bias and the exclusion of non-English studies may have restricted the breadth of evidence considered. These limitations underscore the need for future research to address these gaps and provide a more nuanced understanding of breast cancer-screening dynamics among African immigrant communities.

## 5. Recommendations for Future Research

Based on the findings of this review, we propose several recommendations for researchers to consider when designing culturally tailored breast cancer-screening interventions for African immigrants. As illustrated in Table 5, addressing these recommendations provides the opportunity to further develop and enhance the rigor of studies promoting breast cancer screening among African immigrants.

## 6. Conclusions

Limited research has examined barriers, facilitators, and interventions with respect to breast cancer screening among African immigrant women. Successful promotion of breast cancer screening in this population will require a comprehensive understanding of not only barriers, but also facilitators, which can be leveraged to promote breast cancer-screening engagement. Addressing these factors across all levels of influence according to the social–ecological model in the context of the design of culturally tailored intervention has the potential to maximize the public health impact of breast cancer-screening interventions for African immigrants. Such future work will be imperative for promoting early detection and timely treatment of breast cancer among U.S. African immigrants, as well as reducing breast cancer mortality.

## Figures and Tables

**Figure 1 ijerph-21-01004-f001:**
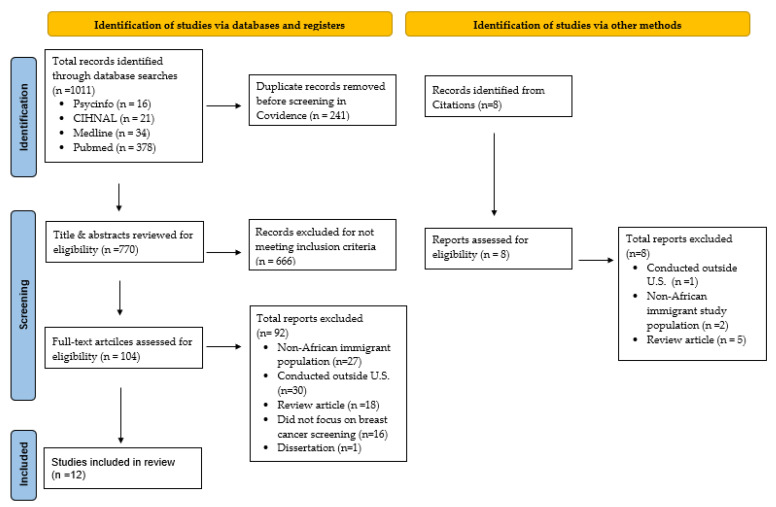
PRISMA diagram illustrating the study selection process.

**Table 1 ijerph-21-01004-t001:** Overview of qualitative studies included in the review (n = 6).

Author and Year	Sample Characteristics	Study Design	Study Purpose	Results	Strengths, Limitations, and Recommendations (Reported by the Authors of the Individual Studies)
Sheppard et al. (2010) [14]	**N**: 20 women **Age**: 21–60 years **Setting**: Washington, DC. **Population**: Western Africa (Nigeria and Ivory Coast) Eastern Africa (Ethiopia) Southern Africa (Zimbabwe) **Number of years in the U.S.:** 3–20 years **Languages:** English	**Methodology:** Descriptive **Method:** Focus group sessions **Data Analysis:** Thematic and content analysis Theoretical model: not stated	To explore African women’s knowledge and attitudes towards breast cancer practices and to identify potential intervention targets	**Barriers:** Limited knowledge of breast cancer screening Lack of insurance Cultural/spiritual beliefs Shame and stigma Privacy and trust Challenging access to the U.S. healthcare system **Facilitators:** Not stated	**Strength:** Use of storytelling by several women to illustrate their points through brief parables **Limitations:** Small sample size Exclusion of non-English-speaking African immigrant women Non-representation of other various subgroups Findings do not include the barriers faced by African women who do belong to the organizations **Recommendations:** To develop culturally tailored breast cancer interventions, additional research will be useful Educating men with assisting with early detection
Al-Amoudi et al. (2015) [25]	**N**: 14 **Age**: 30–69 years old **Setting**: Seattle, Washington **Population**: Muslim Somali women **Number of years in the U.S.:** Recent immigrants (number not stated) **Languages:** English	**Methodology:** Qualitative **Method:** Focus groups (90 min) **Data Analysis:** Thematic and content analysis Theoretical model: not stated	To provide additional insight into the knowledge and beliefs about breast cancer and breast cancer screening among recent Somali immigrant women	**Barriers:** Language Fear of pain Transportation problem Lack of knowledge of breast cancer and breast cancer screening Lack of female or Muslim doctors **Facilitators:** Not stated	**Strength:** All recent Somali immigrants were included in the study **Limitations:** Small study size Limited data collected (demographic not collected) **Recommendations:** Provide breast cancer educational programs to immigrant women to prevent late-stage disease because of lack of screening Ensure female and/or Muslim doctors are available
Saadi et al. (2015) [28]	**N**: 17 **Age**: 27–58 years **Setting**: Chelsea, Massachusetts **Population**: Somali, Bosnian, and Iraqi refugee women (convenience and snowballing sampling) **Number of years in the U.S.:** 2 to 16 years **Languages**: English	**Methodology:** Descriptive **Method:** Semi-structured interview (20-45 min) **Data Analysis:** Thematic and content analysis Theoretical model: grounded theory	To explore Bosnian, Iraqi, and Somali women refugees’ beliefs about preventive care and breast cancer screening to inform future community interventions and best practices	**Barriers:** Fear of pain and diagnosis Modesty Fatalism Inattentiveness to personal health Work and childcare commitments Steep learning curve about navigating and understanding the appointment system **Facilitators:** Access to state insurance coverage Outreach education efforts Women who spoke the same language as participants Shuttles and taxi vouchers Ready access to public transportation Appointment reminders Personal contact with health providers Perceptions of how the American medical infrastructure compared with inadequacies in their home countries Positive attitude toward U.S. health professionals	**Strength:** Easy access to conduct local assessments due to low financial outlay Use of interview **Limitations:** The use of a one-way translation method for the interview questions Lack of an external review of translations and transcriptions that would have strengthened study rigor Interviewers’ type, level, and depth of prior training and experience affected the quantity and depth of data related to interviewing Possible bias was introduced into transcripts by utilizing health center staff to carry out the interviewing, transcription, and translation Possibility that participants may have been influenced by the interviewer’s affiliation with the health center when discussing their perspectives about healthcare providers Use of snowballing and convenience sampling rather than random sampling also reduces the generalizability of the data **Recommendations:** Healthcare equity can be strengthened by ensuring that programs exist that assist refugee women in navigating a complicated and new healthcare system Implementing programs that optimize patient–physician interactions in refugee communities Designing and implementing a message of a health campaign that focuses on the unique needs and experiences of different groups Understanding religious diversity and that it is not universal or monolithic
Pratt et al. (2017) [26]	**N**: 34 **Age**: 18 and over **Setting**: Urban mosque in Minneapolis, Minnesota **Population**: Somalians **Number of years in the U.S.:** <5 years—N = 4 6–10 years—N= 13 >10 years—N =15 Not answered—N = 3 **Languages:** English	**Methodology:** Qualitative **Method:** Focus group (two hours) **Data Analysis:** Thematic Theoretical model: Social constructivist version of grounded theory	To test the acceptability of faith-based messages aimed at ameliorating concerns among Somali women and men that cancer screening conflicts with the religious value of maintaining modesty and the concept of predestination, as well as promoting screening and treatment for cancer	**Barriers:** Fear of the mammography machine Negative experiences with testing in the past Experiences of pain in undergoing screening Faith Modesty Lack of symptoms **Facilitators:** Not stated	**Strength:** Novel, as participants were very engaged even though they were in separate rooms **Limitations:** The impact of actual screening behavior was not tested Samples were self-selected, thus the potential for bias Age or level of religiosity was not explored Small sample size **Recommendations:** To add faith-based messaging to other interventions that focus on improving screening uptake as it could help address health disparity among Somali women Future work should engage families and individuals in conversations about cancer in the Somali immigrant community Incidence of screening should be collected in future works
Ndukwe et al. (2013) [29]	**N**: 38 **Age**: 20–70 years **Setting**: Washington D.C. metropolitan area **Population**: Ghana, Cameroon, Nigeria, Zambia, and Ivory Coast **Number of years in the U.S.:** Not reported **Languages:** English	**Methodology:** Qualitative **Method: ** Key informant interviews (45–60 min) and focus group sessions (45–60 min) Sociodemographic questionnaire (over a 3-month study period) **Data Analysis:** Thematic analysis Theoretical model: not stated	To investigate the knowledge, perceived barriers, and frequency of breast and cervical cancer screening among African immigrant women residing within the Washington D.C. metropolitan area To determine the effects of cultural factors, spiritual beliefs, and familyr influences on breast and cervical cancer screening and to assess whether these beliefs vary with age Supplement the minimal available information regarding the breast- and cervical-cancer outcomes of African-born immigrants to the U.S.	**Barriers:** Lack of awareness Fatalism Stigma/shame Privacy Fear Religious and cultural factors Transportation Cost of insurance Spousal consent/approval **Facilitators:** Reminders from primary-care physicians Cancer-related deaths in the community	**Strength:** Shed light on the cancer outcomes specifically of female African immigrants to the U.S. Collaborating qualitative methods with community organizations to improve health outcomes within this population The findings are critical for researchers, physicians, and public health educators aiming to design culturally appropriate interventions to effectively reduce the prevalence of breast and cervical cancer among female African immigrants living in the U.S. **Limitations:** Small sample size and a limited number of participants’ countries of origin—thus regional-specific in findings Lack of generalization due to only English-speaking participants **Recommendations:** To develop successful outreach to provide more cancer screening among this group to increase early detection and treatment Refrain from categorizing African-born immigrants with AA due to differences in beliefs and practices Utilizing cancer survivors (of similar ethnic descent) in outreach efforts to address the barrier of fatalism Future research should utilize patient navigators to assist women who were diagnosed with breast cancer in adhering to the prescribed treatment plan Utilizing researchers knowledgeable in Biblical scripture to help answer questions cited by participants referencing the Bible Ensure that programs involve churches and religious leaders within the community to become advocates of cancer screening and treatment
Raymond et al. (2014) [27]	**N**: 29 **Age**: 20–65 years **Setting**: Minnesota **Population**: Somali immigrant women **Number of years in the U.S.:** Not stated **Languages:** English	**Methodology:** Qualitative **Method:** Focus group **Data Analysis:** Immersion crystallization Theoretical model: not stated	To gather knowledge to better understand what Somali immigrant women know about cancer, the acceptability of mammograms and pap smears as screening modalities, and any age-based differences in attitudes toward screening, to create a culturally relevant intervention for Somali women living in Minnesota	**Barriers:** Concerns about radiation Modesty and shyness Stigma Pain and embarrassment Lack of accurate knowledge about breast cancer Mistrust of the healthcare system Language barriers Lack of providers from the same religious background **Facilitators:** Increasing the role of women in the community Staying healthy to fulfill the role of childbearing Using religion to encourage health Openness to regular checkups and early identification of health problems	**Strength:** Not stated **Limitations:** Small sample size Findings not generalizable **Recommendations:** Cultural misperceptions and attitudes need to be addressed in developing culturally appropriate interventions to improve screening uptake for Somali women The need to integrate culturally informed beliefs into intervention development, preventive care, and screening promotion

**Table 2 ijerph-21-01004-t002:** Overview of quantitative studies included in the review (n = 6).

Author and Year	Sample Characteristics	Study Design	Study Purpose	Study Outcomes	Strengths, Limitations, and Recommendations (Reported by the Authors of the Individual Studies)
Sheppard et al. (2015) [32]	**N**: 200 women **Age**: 20–60 years **Setting**: Washington, DC metro areas **Population:** West, East, Central Africa, and others **Number of years in the U.S.:** <10 years—N =75 >10 years—N = 112 Missing—N = 13 **Languages:** English	**Methodology:** Cross-sectional **Method:** Self-administered questionnaire (20 min) **Intervention components** N/A Theoretical model: not stated	To examine factors that are associated with higher endorsement of screening To identify areas for more in-depth study	**Barriers:** Lack of insurance Language Marital status **Facilitators:** Not stated **Other findings:** Breast cancer-screening endorsement was higher (81%) among English-speaking women over 40 years, married, insured, living in the U.S. for over 10 years, and having breast cancer knowledge Breast cancer screening rate (88%) was higher in this cohort than other studies (15–61%)	**Strength:** Use of a community-based survey designed and delivered in partnership with a community-based organization Survey that was informed by previous focus groups with African immigrant women Recruitment of an under-represented sample of African women from diverse nationalities The inclusion of items from validated studies The assessment of psychosocial factors (breast cancer knowledge and screening attitudes) that were not previously described in this population The study expands current knowledge about psychosocial factors (not captured in prior studies) **Limitations:** Non-stratification due to small sample size Use of highly insured convenience sample with a high percentage of established immigrants The result cannot be generalized (non-insured or new immigrants) Over-reporting due to self-reported measures The use of single-item outcomes hindered the possibility of assessing internal consistency **Recommendations:** Future studies should consider nuances among diverse women of African origin The need to develop interventions to improve doctor–patient communication via bilingual navigators Development of information materials in different African languages
Raines Milenkov et al. (2020) [33]	**N**: Not stated **Age**: 40 and older **Setting**: Fort Worth, Texas **Population**: Central African Region (Democratic Republic of Congo, Rwanda, Burundi, Tanzania, and Uganda) Somalia/Kenya Arabic (Sudan, Iraq, Syria, Egypt, Jordan) Other (Afghanistan, Angola, Chad, Eritrea, Ethiopia, Liberia, and Senegal) **Number of years in the U.S.:** Not stated **Languages:** English	**Methodology:** Cross-sectional **Method:** Education program (video, presentation, and anatomic model) Theoretical model: not stated	To assess differences in uptake of cervical, breast, liver, and colorectal screens across six cultural groups	**Barriers:** Lack of transportation Trauma Health literacy Education levels Lack of culturally congruent providers Complex nature of the U.S. healthcare system Interpretation Scheduling and lack of appointment reminders **Facilitators:** Not stated **Other findings:** The uptake for breast cancer screening was 50% or more in all cultural groups except one, the Central African region (35%) Mammograms were the most accepted cancer screen among most refugees (54%), except the Central African group (35%) The odds of participants from the Central African region group accepting a mammogram were 63% lower than the odds of a mammogram being accepted by members of the Myanmar group (POR = 0.37, 95% CI = 0.18–0.73, *p*-value = 0.01). More women received a mammogram (54%) There is evidence that uptake in screenings varies among ethnic groups, even with a culturally and linguistically appropriate outreach and education strategy	**Strength:** Focused on never-screened and not up-to-date participants **Limitations:** Lack of generalizability of study findings due to the Texas healthcare environment Self-reporting of baseline assessment data The possible effect of the community health workers’ characteristics could have prevented members from willingly enrolling in breast cancer screening A smaller sample size in the Arabic-speaking group reflects delayed outreach into this population **Recommendations:** To meet national goals, detect cancer early, and save lives, there needs to be a culturally and linguistically appropriate outreach and education to diverse immigrant groups Focus on alternate education delivery methods, such as online or video-based Future studies should consider different study settings such as clinics and community locations Understanding diversity within refugee communities and adapting to their specific cultural and linguistic needs with information, education, and outreach will help reduce the inequality gap
Pratt et al., (2020) [35]	**N**: 30 **Age**: 30–70 years old **Setting**: Local Mosque-Minneapolis, Minnesota **Population**: Somali Muslim Women **Number of years in the U.S.:** 1–23 years **Languages:** English	**Methodology:** Cross-sectional **Method:** Religiously tailored workshops (I week) Theoretical model: Social–ecological approaches	To test the feasibility and impact of religiously tailored workshops in helping to promote breast and cervical cancer screening within the Somali Muslim community	**Barriers:** Modesty Faith Pre-destination Access to preventive health care **Facilitators:** Not stated **Other findings:** All women reported that the workshop provided beneficial information That they are more likely to receive breast cancer screening Positive baseline result Fewer women believed that screening may cause harm in post-workshop surveys than in pre-workshop surveys	**Strength:** The use of an Imam to clarify Islamic understandings of relevant issues had the potential to utter religiously attributed beliefs that were acting as a barrier to breast cancer screening **Limitations:** Bias, as sampled women were currently in breast cancer screening and thus were more open to screening No verification of screening due to self-reporting Potential desirability bias, since the workshop was facilitated by a religious leader Small size sample **Recommendations:** To test the message on Somali women who are not up-to-date with screening To test the intervention on a broader range of religiously attributed barrier beliefs to assess the extent of participant religiosity and the role of both self- and collective-efficacy on screening behavior
Piwowarczyk et al. (2012) [34]	**N**: 120 **Age**: 25–64 years **Setting**: Community sites- Greater Boston and New Hampshire **Population**: Somali and Congolese women **Number of years in the U.S.:** 0–19 years **Languages:** English	**Methodology:** Cross-sectional **Method:** Pretest survey (30 min) One session group workshop–DVD (2 h) Post-test survey Theoretical model: not stated	To evaluate the UJAMBO program addressing the impact on participants’ knowledge of health services and their intentions to use these services	**Barriers:** Lack of knowledge of mammography **Facilitators:** Not stated **Other Findings:** Significant increase in knowledge as to the purposes of these services An increase in the intent to pursue the services Significant improvement in knowledge of mammograms at post-test Significant improvement in intentions to receive mammograms	**Strength:** Not stated **Limitations:** Possibility of social desirability as a reason for changes in reported intentionality Findings were limited to pre-post analyses subject to testing effects Lack of behavioral outcome data and other threats to validity Pre- and post-tests were administered too close together Outcome data were solely based on self-report, subject to social desirability biases Lack of generalizability, as the sample had women from only two African immigrant countries with women linked to the CBOs serving those communities **Recommendation:** Conduct larger-scale research with more rigorous evaluation methods
Harcourt et al. (2013) [31]	**N**: 112 **Age**: 40 years and above **Setting**: Minneapolis and St. Paul (participants’ home) **Population**: Somali Other African women **Number of years in the U.S.:** <5 years—N = 44 >5 years—N = 68 **Languages**: English	**Methodology**: Cross-sectional (secondary data) **Method:** Survey Theoretical model: Healthcare access and utilization Behavioral model for vulnerable populations	To determine the rates of participation in breast- and cervical-cancer screening among age-eligible female African immigrants and to examine barriers associated with these cancer screening procedures	**Barriers:** Difficulty with healthcare access **Facilitators:** Not stated **Other findings:** African immigrant women in Minneapolis and St. Paul have low breast- and cervical-cancer screening rates Ethnicity (Somali vs. other African immigrant groups) and duration of residence in the U.S. (≤5 years vs. >5 years) were significantly related to ever having had a mammogram Somali women were 5 times more likely than other African women to ever having had a mammogram (odds ratio = 5.02, 95% CI = 1.72–14.68, *p* = 0.003)	**Strength:** Use of a conceptual model framework to explore factors impacting breast cancer-screening behavior among African immigrant women **Limitations:** Limited sample size Restriction of the data to urban areas in Minneapolis and St. Paul with the largest concentration of African immigrant families Secondary data analysis Self-reported breast cancer screening **Recommendations:** Community-based educational interventions should focus on the need for screening among all immigrant women in a culturally sensitive manner Future research should explore the impact of acculturation on cancer screening among recent immigrant women
Adegboyega et al. (2022) [30]	**N**: 59 **Age**: 40 years and over **Setting**: Lexington and surrounding cities in Kentucky **Population**: Sub-Saharan women (Nigeria, Cameroon, and Congo) **Number of years in the U.S.:** Not stated **Languages:** English	**Methodology:** Cross-sectional **Method:** Survey (20 min) Theoretical model: Social cognitive theory (SCT)	To evaluate the relationships between beliefs (religiosity, fatalism, temporal orientation, and acculturation) and cervical-, breast-, and colorectal-cancer screening behaviors among African Americans and Sub-Saharan African immigrants	**Barriers:** Fatalism, religiosity, belief **Facilitators:** Not stated **Other Findings:** Religiosity, cancer fatalism, future orientation, and acculturation, were not associated with mammography screening Mammogram uptake increased with age	**Strength**: Not stated **Limitations:** The study design was a cross-sectional research design, which does not reflect causation Use of convenience and non-probability samples Modest sample size, which was further restricted for each model based on those who were appropriate for the screening modality Self-reported before completion of screening, which may be subject to recall or social desirability bias **Recommendations:** To promote preventive screening, health researchers should consider health temporal orientation and other beliefs in the design of interventions for Black populations Increase breast cancer prevention and awareness efforts among younger individuals Larger-scale, powered studies with increased inclusion of participants and robust sampling

**Table 3 ijerph-21-01004-t003:** Barriers to breast cancer screening according to level of influence based on the Social Ecological Model [23,24].

Level of Influence	Barriers	Total N	Articles Citing this Barrier
**Intrapersonal**: Individual characteristics that influence breast cancer screening, including attitudes, beliefs, knowledge, and personality traits	Lack of awareness/limited knowledge about breast cancer screening.	7	[14,25,26,28,29,32,34]
	Fear of pain during breast cancer screening procedures and/or the disease following breast cancer diagnosis.	4	[25,27,28,29]
	Inability to speak the English language.	4	[14,25,27,32]
	Limited transportation to breast cancer-screening facilities.	3	[25,29,33]
	Competing priorities (i.e., work and childcare commitments).	2	[28,29]
	Health concerns related to mammography.	2	[26,27]
	Costs associated with breast cancer screenings.	1	[29]
	Past negative experience with breast cancer screening.	1	[26]
**Interpersonal**: Primary social groups and cultural influences on breast cancer screening; includes family, friends, peers, and cultural norms	Sociocultural norms related to modesty and privacy.	7	[14,25,26,27,28,29,35]
	Spiritual beliefs and religious practices (faith and fatalism).	6	[14,26,28,29,30,35]
	Lack of social support.	1	[32]
**Community:** Social structures and policies that influence breast cancer screening	Lack of culturally congruent providers.	3	[14,25,27]
	The complex nature of the U.S. healthcare system.	2	[14,28]
	Lack of access to breast cancer-screening providers.	2	[31,32]

**Table 4 ijerph-21-01004-t004:** Facilitators of breast cancer screening according to level of influence based on the Social Ecological Model [23,24].

Level of Influence	Facilitators	Total N	Articles Citing this Barrier
**Intrapersonal**: Individual characteristics that influence breast cancer screening	Previous breast cancer-screening experience	1	[29]
	Health insurance	1	[28]
**Interpersonal**: Primary social groups and cultural influences on breast cancer screening; includes family, friends, peers, and cultural norms	Observing other community members’ experiences with breast cancer	1	[29]
**Community:** Social structures and policies that influence breast cancer screening	Appointment reminders and scheduling assistance	2	[28,29]
	Interpreters with the same culture	1	[28]
	Transportation to breast cancer screenings	1	[28]
	Patient navigators	1	[28]

**Table 5 ijerph-21-01004-t005:** Recommendation for future research.

Recommendation	Description
**1.** Recognize that African immigrants are not monolithic and have different healthcare experiences and delivery preferences based on their country of origin and prevalent sociocultural norms	Researchers should acknowledge the heterogeneity that exists among African countries and the diverse healthcare experiences among women from these countries Researchers should understand the varied experiences and barriers related to breast cancer among distinct African immigrant groups
**2.** Conduct additional research examining facilitators and/or motivators for engaging in breast cancer screening	Limited research has examined facilitators or motivators of breast cancer screening among African immigrants. Such factors represent key leverage points for researchers to include in the design of culturally tailored breast cancer-screening interventions Authors conducting future studies are encouraged to explicitly examine facilitators identified in our review to further elucidate the role of these facilitators in promoting breast cancer screenings, while also exploring other potential facilitators not previously reported
**3.** Conduct additional quantitative studies examining barriers to and facilitators for breast cancer screening	The majority of research examining barriers to and facilitators of breast cancer screening among African immigrant women has been qualitative. It is vital for further research to confirm and build upon qualitative insights, crucial for tailoring effective interventions in breast cancer screening
**4.** Conduct additional, rigorously designed, theory-driven intervention studies promoting breast cancer screening among African immigrants	We advocate for more extensive exploration and testing of culturally sensitive breast cancer-screening interventions in larger-scale and more rigorous trials Trials should not only assess psychosocial constructs targeted by the intervention, but also include the evaluation of the actual prevalence of breast cancer-screening engagement following the intervention We also recommend that interventions draw upon established theories and models such as health-behavior theories (e.g., the Health Belief Model and Theory of Planned Behavior) when designing breast cancer interventions
**5.** Ensure collaboration between researchers and health agencies.	Future research should include collaborations between researchers and public health agencies to design and implement culturally tailored breast cancer-screening interventions that could serve as a significant platform for increasing BC-screening uptake within this demographic For example, ref. [33] collaborated with community health workers and leaders to deliver culturally and linguistically suitable cancer-prevention education to refugees from six cultural backgrounds in the U.S. The findings indicated a significant increase in breast cancer-screening rates, with most cultural groups experiencing a rise of 50% or more, including Somali women

## Data Availability

No new data were created or analyzed in this study. Data sharing is not applicable to this article.

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
