# Peer review of "Breast Cancer Screening among African Immigrants in the United States: An Integrative Review of Barriers, Facilitators, and Interventions"

_ijerph, 2024, doi:10.3390/ijerph21081004_

Round 1
Reviewer 1 Report
Comments and Suggestions for Authors
This manuscript is a interesting analysis of barriers and facilitators of Breast cancer screening among African immigrants in the United States. The manuscript is well organized and respects the recommended structure for a systematic review including Prisma criteria and Prospero registration.
The methodology and results are clearly presented and the conclusions are supported by the text.
My suggerstions for improving of the text are as follows>
1. Please add a table with the PICOS criteria for study selection
2. Please reedit the table with the study caractheristics - the last collum needs to be wider to ensure better readability - as it stands it lengthens the table on many pages
3. I would have liked to see disscused specific differences between African emigrants and the rest of the US population and how they are compared to the African-American population - specific details about housing conditions, levels of education, levels of income and unemployment, health insurance and differences in psiho-social structures (religious aspects, community aspects, past trauma - many of them originate in poor and war-throrn countries, female vs male education levels).
4 Minor typos should be adressed.
In my opinion this manuscript can be pubished after minor changes.
Reviewer 2 Report
Comments and Suggestions for Authors
This article describes an integrative review of barriers, facilitators, and interventions of breast cancer screening among African immigrants in the United States. The authors contend that Black women in the United States experience worse cancer-related outcomes compared to women of other racial and ethnic groups. Moreover, authors assert that prior research does not acknowledge the heterogeneity that exists among Black women in the U.S., including whether women are born in the U.S or are immigrants or the unique barriers to healthcare that Black immigrants in the U.S. face. This article seeks to address a gap in the literature by investigating the breast cancer screening experiences among African immigrants in the United States.
The review is as follows:
1. The wording is unclear in this sentence: “Observing other women experiences breast cancer was identified as an interpersonal facilitator…” (lines 24-25).
2. Figure 1 in the Methods section is difficult to read. Consider increasing the font.
3. In the first box in Future 1 (Records identified from), it will be helpful for authors to include the total sum (n) of all records identified.
4. Good application if the Socioecological Model to barriers to breast cancer screening (Table 3). Include a citation for the Socioecological Model to accompany the table.
5. Check for capitalized ‘s’ in “Intrapersonal: Individual characteristics that influence breast cancer Screening” (Table 3).
6. Check for capitalized ‘s’ in “Intrapersonal: Individual characteristics that influence breast cancer Screening” (Table 4).
7. Good inclusion of recommendations for future research.
Overall, this is a unique, comprehensive, compelling paper on a pertinent topic, It is well-written and interesting to read. The paper can make a distinct contribution to the existing literature. Addressing some minor areas in the paper as suggested in this review can help make this paper suitable for publication.
Comments on the Quality of English LanguageThere are minor items to address related to grammar. Overall, the quality of the English language is fine.
